# The Quest for Non-Invasive Diagnosis: A Review of Liquid Biopsy in Glioblastoma

**DOI:** 10.3390/cancers17162700

**Published:** 2025-08-19

**Authors:** Maria George Elias, Harry Hadjiyiannis, Fatemeh Vafaee, Kieran F. Scott, Paul de Souza, Therese M. Becker, Shadma Fatima

**Affiliations:** 1Medical Oncology, Ingham Institute for Applied Medical Research, Liverpool, NSW 2170, Australia; m.elias3@westernsydney.edu.au (M.G.E.); 19837765@student.westernsydney.edu.au (H.H.); paul.desouza@sydney.edu.au (P.d.S.); 2School of Science, Western Sydney University, Locked Bag 1797, Penrith South DC, Sydney, NSW 2751, Australia; 3School of Medicine, Western Sydney University, Locked Bag 1797, Penrith South DC, Sydney, NSW 2751, Australia; kieran.scott@westernsydney.edu.au (K.F.S.); therese.becker@inghaminstitute.org.au (T.M.B.); 4School of Biotechnology and Biomolecular Science, University of New South Wales, Sydney, NSW 2052, Australia; f.vafaee@unsw.edu.au; 5UNSW AI Institute, University of New South Wales, Sydney, NSW 2052, Australia; 6Nepean Clinical School, The University of Sydney, Sydney, NSW 2747, Australia; 7Centre for Circulating Tumour Cell Diagnostics and Research, Ingham Institute for Applied Medical Research, Liverpool, NSW 2170, Australia; 8South Western Sydney Clinical School, University of New South Wales, Liverpool, NSW 2170, Australia

**Keywords:** glioblastoma multiforme, liquid biopsies, biomarkers

## Abstract

Liquid biopsy is gaining momentum as a non-invasive tool for diagnosing and monitoring the most aggressive brain cancer, glioblastoma (GBM). Unlike brain biopsies, which require invasive surgery, liquid biopsies use biofluids such as blood, urine or cerebrospinal fluid to detect tumour-derived molecules. This review discusses the recent advances and inherent potential of liquid biopsy derived tumour DNA, RNA, proteins, nucleosomes and extracellular vesicles, to improve early detection and enable real-time disease monitoring. Despite promise, challenges remain due to the blood–brain barrier, low biomarker abundance, and inconsistent methods. Emerging computational approaches with AI/ML tools are helping overcome these issues by extracting insights from complex data and integrating multi-analyte profiles that may eventually surpass the limitations of traditional diagnostic methods especially for brain tumours.

## 1. Introduction

Glioblastoma Multiforme (GBM) is the most common and aggressive primary brain tumour, with a median survival of around 15 months despite intensive treatments including surgery, radiation, and chemotherapy [1]. Its highly invasive nature, resistance to therapy, and recurrence present ongoing treatment challenges. GBM is also marked by substantial molecular heterogeneity, including gene mutations such as IDH-R132H, H3-K27M, and EGFR (EGFRvIII) amplification complicating treatment approaches and prognosis. Moreover, the blood–brain barrier (BBB) further limits the effectiveness of conventional therapies and monitoring methods [2,3,4].

The diagnosis of GBM typically includes neuroimaging, followed by histopathological and molecular analysis for confirmation [5,6,7]. However, both imaging and tissue-based methods have limitations, such as distinguishing between relapse and pseudo progression, which are changes related to treatments that simulate recurrence of GBM [8]. These challenges underscore the need for alternative, non-invasive diagnostic tools, such as liquid biopsy, which has surfaced as a promising approach to detect early recurrence for GBM patients, principally through blood tests, urine, or cerebrospinal fluid (CSF), thereby providing real-time insights into tumour dynamics and treatment response [9,10]. Liquid biopsies comprise the detection and quantification of tumour-related content in biofluids that is released by tumours or in response to the presence of tumour, such as circulating tumour cells (CTCs), circulating tumour DNA (ctDNA), circulating tumour microRNA (miRNA), extracellular vesicles (EVs) and nucleosomes, all of which can provide critical insights into GBM progression and recurrence [11].

Being minimally or non-invasive, liquid biopsy offers a valuable alternative to invasive tissue biopsies for detecting early recurrence, monitoring minimal residual disease (MRD), and guiding treatment strategies in GBM. While the BBB hinders the amount of tumour-derived entities that may be found in body fluids, ongoing research aims to increase sensitivity and specificity of liquid biopsy tests, to enhance their promise as a tool for advancing clinical management of GBM. This review will explore recent advancements in liquid biopsy technologies and their potential for enhancing the diagnosis and treatment of GBM.

## 2. Circulating Biomarkers of Glioblastoma from Liquid Biopsies

### 2.1. Circulating Tumour Cells

CTCs are individual cells or cell clusters that are released from primary or metastatic solid tumours into bodily fluids. When disseminated into the bloodstream and other body fluids, they may contribute to the metastastic process and can be used as biomarkers for cancer detection, prognosis, and therapeutic monitoring [9,12]. The analysis of CTCs has emerged as a potential tool for monitoring MRD in cancer patients [13,14,15]. Higher levels of CTCs are directly related to the severity and spread of multiple cancers [16]. Key methods for CTC enumeration, which quantify CTC numbers, include sensitive assays like CellSearch^®^ (Menarini Silicon Biosystems, Bologna, Italy), which detects as few as one CTC per 10 mL of blood. However, CellSearch^®^ uses antibody-mediated capture, mainly targeting epithelial cell adhesion molecule (EpCAM) on CTC membrane surfaces using a positive selection method [17,18,19,20,21].

EpCAM is not expressed in GBM tumours due to their predominantly mesenchymal phenotype [12,22]. Hence, alternative techniques for the isolation of CTCs in GBM patients have been explored (Table 1). One such method is negative selection, which enriches CTCs by eliminating leukocytes from whole blood by means of antibodies against leukocyte antigens [9,19]. Several different techniques of CTC isolation have been used which are dependent on the physical properties of CTCs [23,24]. These include density-gradient centrifugation [12,25], spiral microfluidic technology [26,27] and chip technology [23,24]. Although promising, it is important to highlight the extreme rarity of CTCs in peripheral circulation in GBM patients. Reported detection rates vary widely, with some studies identifying CTCs in as few as 20-40% of GBM patients, depending on the sensitivity of the detection method and sample volume and timing (20.6–77% of patients with detectable CTCs, Table 1). Along with the difficulty of isolating CTCs from peripheral blood, this rarity underscores the need for highly sensitive and specific detection technologies to validate their clinical utility as reliable biomarkers. It also further complicates standardisation efforts and highlights the need for optimisation and harmonisation of isolation protocols.

CTC analysis methods for GBM, other than enumeration, include molecular profiling (e.g., next generation sequencing (NGS), Reverse Transcriptase-PCR (RT-PCR)), immunophenotyping using GBM-specific markers (such as GFAP, Nestin, Sox2, EGFR, GLAST) [21,28], single-cell sequencing [25], and epigenetic profiling [29], with CSF-based analysis often offering higher sensitivity than blood [30,31,32]. However, it is essential to acknowledge that CSF provides a higher concentration of brain-derived biomarkers unlike blood or other fluids because of its direct contact with the brain’s extracellular space, allowing for more accurate detection. Notably, markers like SOX2 and GFAP are detected via protein expression using immunocytochemistry or flow cytometry, whereas transcript analysis methods such as RT-PCR target mRNA expression, distinguishing these approaches technically. These methods enable the detection of clinically relevant alterations, such as IDH1 mutations and MGMT promoter hypermethylation, which are important for prognosis and for guiding targeted therapy decisions in GBM treatment [19]. 

Monitoring CTC kinetics over time can provide insights into the effectiveness of treatment and the risk of disease recurrence. Changes in CTC count, phenotype or genetic profile can act as early indicators of therapeutic response of tumour progression [21,22,25,33,34] (Table 1).

**Table 1 cancers-17-02700-t001:** CTC isolation and analysis in patients with GBM.

Isolation Method	Patient Numbers	Markers Used to Verify GBM Origin	Findings	References and Publication Date
CTC-iChip microfluidic technology. Leukocyte depletion using magnetically tagged anti-CD45 and anti-CD16 antibodies.	33	SOX-2, EGFR, c-MET, A2B5 tubulin *β*-3	Isolated CTCs from peripheral blood of 39% (13 of 33) GBM patients. Greater CTC counts in patients with progressive disease relative to stable disease. CTCs have a mesenchymal phenotype.	[22] (2014)
Density gradient centrifugation	141	Single-cell genomics for common GBM mutations, GFAP staining, tumour specific anomalies like amplification of EGFR gene and gains and losses in chromosomes 7 and 10 genomic regions by chromosomal and array CGH on whole genome amplification.	Isolated CTCs from 20.6% (29/141) of patients. A convenient diagnostic tool for identifying patients with extracranial tumour cell spread and indicates that CTCs could be used to monitor the progression of glioblastoma.	[34] (2014)
Density gradient centrifugation using the OncoQuick^®^ (Greiner Bio-One, Frickenhausen, Germany), system.	11	Telomerase-based test was used to identify CTCs.	CTCs detected in 72% (8 of 11) of patients before radiotherapy but dropped to 8% (1 of 8) in post-radiotherapy patients. This suggests that CTCs may be useful to monitor the progression of cancers before and after therapies.	[25] (2014)
Immunoaffinity-based methods. CTC separation with a matrix and negative depletion of white blood cells using immunomagnetic beads.	31	Polyploidy chromosome-8-positive detection was employed as a positive measure for CTCs using subtraction enrichment and immunostaining-fluorescence in situ hybridization (SE-iFISH), in addition to GFAP-positive or GFAP-negative cells and CD45-negative cells grading to confirm cell origin.	CTCs were detected in 77% (24 of 31) of GBM patients. Monitoring treatment using CTCs was slightly better than MRI in distinguishing radionecrosis from recurrence of glioma. CTCs can dynamically monitor the microenvironment of gliomas which is a significant complement to radiographic imaging.	[35] (2016)
Immunotargeted enrichment of MSP and MCAM expressing cells.	13	CTCs were isolated based on cell surface MSP and MCAM and identified by probing for GLAST and/or GFAP expression.	≥1 CTCs were detected in 69% of patients (9/13), using the combination of 2 isolation and 2 identification markers increased CTC detection.	[21] (2020)
Immunopheno-typing. CTCs were isolated by size separation using a Parsortix^®^ (ANGLE plc, Surrey, UK), microfluidic cassette.	13	CTCs were isolated based on no expression of CD45, while expressing EGFR, Ki67, and EB1 microtubule associated protein. For confirmation, the GBM CTC clusters and a biopsy from the primary tumour of the patient were stained with GBM marker SOX-2.	CTC clusters identified in 53.8% of 13 GBM patients. GBM markers validated that multicellular CTC clusters can be formed and pass the BBB in patients with GBM to reach peripheral circulation and be used for monitoring.	[28] (2018)
Spiral microfluidic technology	20	CTCs exhibited characteristic molecular features of GBM, such as EGFR amplification and mutations in TP53 and IDH1.	The study found that CTCs could be isolated from both early- and late-stage GBM patients, highlighting the potential of this technique for non-invasive monitoring of tumour progression.	[27] (2021)
CTC Subtraction enrichment/depletion with magnetic immunoaffinity beads and immunostaining-FISH	22	Detection of CTCs to differentiate between treatment-induced necrosis and tumour recurrence in brain gliomas. CTC detection outperformed both DSC-MRP and MET-PET in diagnostic accuracy. Additionally, it showed potential for predicting recurrence in one patient.	The mean CTC count was significantly higher in patients with tumour recurrence (6.10 ± 3.28) compared to those with treatment necrosis (1.08 ± 2.54). A threshold CTC count of 2 provided 100% sensitivity and 91.2% specificity (AUC = 0.933) for identifying tumour recurrence. CTC detection could be a valuable tool for distinguishing tumour recurrence from necrosis, warranting further validation in larger clinical studies.	[36] (2021)

These studies demonstrate the potential of CTCs as a tool for cancer diagnosis and prognosis; however, there are still challenges that need to be addressed: (i) Low abundance: CTCs are present in very low numbers in the bloodstream, typically being 1 CTC in 10^9^ normal blood cells per 10 mL of blood [9,18]. Low abundance can lead to false-negative results and reduce the sensitivity of CTC-based assays. (ii) Heterogeneity: CTCs show intra- and inter-patient heterogeneity and may differ in morphology, phenotype, and genetic profile. This heterogeneity complicates the identification and characterisation of CTCs, and it may require the use of multiple markers or assays to detect different subsets of CTCs [37,38]. (iii) Technical variability: The isolation and analysis of CTCs require specialised equipment and protocols, and it can affect the accuracy and reproducibility of CTC-based assays. The choice of isolation and analysis method, as well as the handling and storage of blood samples, can all impact the results of CTC assays [39,40]. (iv) Lack of standardisation: There is currently no standardised protocol for the isolation and analysis of CTCs, and different studies may use different methods or markers to detect CTCs. This lack of standardisation can make it difficult to compare results across studies and to establish the clinical utility of CTC-based assays [41]. (v) Cost and accessibility: CTC-based assays can be costly and require specialised equipment and expertise, which may limit their availability and accessibility in a clinical setting [42].

Overcoming these barriers will require continued innovation in sensitive and specific CTC isolation and analysis techniques, along with greater collaboration between researchers and clinicians in establishing the clinical utility of this approach. By integrating CTC-based assays with other clinical and pathological parameters, and rigorously validating these methods in clinical trials, the clinical utility and potential for earlier accurate cancer detection and monitoring tumour progression may be determined.

### 2.2. Circulating Tumour DNA

Cell-free DNA (cfDNA) are short DNA fragments of 140-170 base pairs (bp) present in body fluids. In cancer patients, a portion of cfDNA originates from tumour cells following apoptosis or necrosis, referred to as circulating tumour DNA (ctDNA) [9,10,18,43,44]. Representing a promising tool for non-invasive cancer diagnosis, particularly in GBM, ctDNA consists of shorter DNA fragments (<145 bp) that carry the genetic and epigenetic signatures of the originating tumour [12,18]. A number of studies have validated that ctDNA reflects the genetic profile of the original tumour, showing uniformity with its mutational profile and that of the corresponding tumour tissue across various cancers, including GBM [9,45,46,47], thereby offering insights into tumour genetics and treatment response (see Table 2). Gene mutations in, EGFR, PTEN, IDH1, TP53, FGFR2, and ERBB2 have been detected in CSF-derived ctDNA of GBM patients [48]. Key detection methods include assessing ctDNA levels and known point mutations using a polymerase chain reaction (PCR)-based method such as digital droplet PCR (ddPCR) or next-generation sequencing (NGS) and whole-genome sequencing (WGS), methods that detect novel or unknown mutations [49,50,51]. ddPCR is particularly sensitive for identifying low-frequency mutations such as IDH1, TERT promoter mutations, and EGFRvIII in plasma samples [51]. It is important to emphasise that mutation detection in cf/ctDNA, in particular the detection of rare alleles, may be affected by biological background noise. Clonal hematopoiesis (CH) is an event triggered by the accumulation of somatic mutations during the development of hemopoietic cells followed by the outgrowth of genetically more diverse clones. This is a normal process with increasing accumulation of clones with age. Critically, if certain variants in DNA damage response proteins including ATRX and TP53 arise, outgrowth of clones with these mutations (and others in the same cell) is favoured due to impaired DNA repair [52,53]. This means mutations that are generally associated with cancer may also accumulate and be detectable in the non-cancerous blood cell derived component of cfDNA of healthy, especially older individuals. Thus, CH affects the interpretation of liquid biopsy mutation data for rare alleles in GBM. This must be kept in mind and should be controlled for by independent methods of determining ctDNA proportion or the presence of CH mutations in the immune cell population of a patient sample. NGS enables comprehensive genomic profiling, detecting a broad spectrum of mutations and copy number alterations [54]. Methylation-specific PCR targets epigenetic modifications, like MGMT promoter methylation, which are relevant for GBM prognosis and therapy selection [50]. 

ctDNA detection in GBM patients may be hampered by the tumour location and the BBB, which decreases the yield of ctDNA in peripheral circulation. Employing highly sensitive molecular technologies may help overcome these obstacles, as described in Table 2. It is noteworthy that the baseline level of cfDNA is greater in serum compared to plasma, which may be a result of contamination with DNA that is released from lysed immune cells; thus, plasma samples are preferred for studying ctDNA [49,50,51]. While plasma ctDNA alone may not yet be sufficiently sensitive for reliable GBM diagnosis or monitoring, detectable levels offer promise for non-invasive molecular profiling and could help identify actionable mutations and resistance mechanisms, potentially improving treatment strategies. Studies have shown that ctDNA can be detected in a subset of GBM patients, though plasma detection rates are generally lower than in other cancers, likely due to the BBB limiting ctDNA release into the bloodstream [55]. Piccioni et al. found ctDNA mutations in the blood of 50% of brain tumour patients, with 55% of these being GBM [56]. However, dependent on patient cohort and cfDNA extraction method, 75.0% sensitivity (95% CI: 64.1–84.0%) and 88.7% specificity (95% CI: 77.0–95.7%) has been achieved for detection of the IDH1 mutation R132H, and negativity for this mutation is a key classifier of GBM [57,58].

EGFRvIII ctDNA is also detectable in plasma and can serve as a dynamic biomarker for preoperative and postoperative assessment, as well as for monitoring disease recurrence in GBM patients [59]. Zill et al. used next-generation sequencing of plasma ctDNA in 107 GBM patients to identify actionable mutations, supporting the use of ctDNA sequencing for personalised cancer treatment [60]. Liu et al. and Miller et al. further demonstrated that WGS of CSF ctDNA achieves high detection rates and can identify clinically relevant mutations, reinforcing its utility as a minimally invasive biomarker for GBM monitoring [61,62].

Furthermore, studies have demonstrated that CSF is a more reliable source of ctDNA in GBM patients compared to plasma, confirming the restrictions imposed by the BBB in regard to ctDNA release [63,64,65]. For instance, ctDNA was detected in 61% of CSF samples compared to 37% of serum samples from 19 GBM patients using methylation-specific PCR [66], while nested PCR allowed for detection in 92% of CSF samples compared to only 8% in plasma samples from 38 GBM patients [67]. Another study using WGS on CSF samples from 11 GBM patients demonstrated a 100% ctDNA detection rate [68]. These findings highlight the superior sensitivity of CSF-based liquid biopsies, likely due to the proximity of CSF to the brain tumour and the reduced interference from background cfDNA found in blood [9,12,18,69]. However, CSF collection via lumbar puncture is invasive and may not be suitable for all patients [18]. Fluids such as nasal secretions or saliva may provide an alternative to CSF, offering a less invasive means to access brain-derived biomarkers but need further investigation. 

Urine has also emerged as a non-invasive source of ctDNA for GBM diagnosis, offering the advantage of easy and repeatable collection throughout treatment [10]. Following renal filtration, ctDNA can be detected as transrenal DNA (trDNA), enabling molecular diagnostics through urinary DNA analysis [10,70]. Several urinary biomarkers—including matrix metalloproteinases (MMP-2, MMP-9), neutrophil gelatinase-associated lipocalin (NGAL), and vascular endothelial growth factor (VEGF)—have demonstrated high sensitivity (95.2%), specificity (95.7%), and accuracy (92.5%) in detecting primary brain tumours, including GBM (see Table 2). Notably, elevated levels of these urinary markers in GBM patients have been shown to correlate with their expression in tumour tissue and in CSF (for MMP-9 expression) [70,71,72,73]. In addition, urine-based detection of ctDNA epigenetic alterations, such as DNA methylation, offers further potential. Methylation-specific PCR enables the quantification of tumour-specific methylation, such as O6-methylguanine-DNA methyltransferase (MGMT) promoter methylation [74,75], a well-established predictor of GBM patient response to alkylating agents like temozolomide (TMZ) [76,77,78]. TMZ, a DNA alkylating agent, adds methyl groups primarily to the O6 position of guanine, causing lethal DNA damage. MGMT repairs this damage, so its promoter methylation or mutation, which silences MGMT, increases tumour sensitivity to TMZ. Importantly, TMZ treatment also induces substantial changes in the tumour DNA landscape, including increased mutational burden and epigenetic remodelling. These therapy-induced alterations can affect the detection and interpretation of ctDNA and methylation biomarkers, potentially complicating longitudinal monitoring. Therefore, understanding TMZ’s impact on liquid biopsy profiles is essential for accurately assessing treatment response and resistance.

In conclusion, ctDNA analysis has shown significant promise as a tool for detecting and monitoring genetic alterations in brain cancers, although some challenges still need to be addressed. These include variability in ctDNA levels depending on the stage and anatomical location of cancer, as well as its rapid degradation in circulation and sensitivity to pre-analytical factors, which can complicate collection and analysis. Overall, detection of ctDNA in CSF and plasma provides valuable insights into tumour evolution, resistance mechanisms, and potential therapeutic targets. Further advancements in ctDNA isolation, sequencing methods, and clinical validation will be crucial to unlock its full potential for early diagnosis, treatment monitoring, and personalised cancer management.

**Table 2 cancers-17-02700-t002:** ctDNA isolation and analysis in patients with GBM.

Isolation Method	Patient Numbers	Minium Input DNA	Markers Used to Verify GBM Origin	Findings	References and Publication Date
Guardant360^®^ (Guardant Health, Palo Alto, CA, USA) and digital NGS	33	Not specified (from ~10 mL plasma)	NGS targeting 54 cancer related genes, including assessments for copy number variants in *EGFR*, *ERBB2,* and *MET.*	Of the patients diagnosed with GBM, 73% had unaltered ctDNA, 24% had one alteration and 3% had two or more alterations. SNV detection > 85% sensitivity; > 99.99% specificity	[55] (2016)
Guardant360^®^ and digital NGS	222	Not specified	Single-nucleotide variants were detected in 61 genes, with amplifications detected in *ERBB2*, *MET*, *EGFR*.	ctDNA mutations were detected in blood samples from 55% of GBM patients.	[56] (2019)
Illustra triplePrep Kit (GE healthcare BioSciences Corp, Piscataway, NJ, USA) and WGS	13	200 µL plasma (~6 ng cfDNA)	EGFRvIII mutation characterised by a deletion of exons 2 through 7.	EGFRvIII mutant DNA detected in the plasma of GBM patients, with its presence correlating with the mutation in tumour tissue. Limit of detection (LoD) ~0.01% mutant allele frequency	[59] (2013)
DNA extraction and NGS	107	Not specified (from ~10 mL plasma)	Tumour-specific mutations such as EGFRvIII	ctDNA detection rate was 51% Genomic alteration in the ctDNA of patients highlight the potential of guiding personalised cancer treatment. LoD ~0.1% allele frequency	[60] (2018)
DNA extraction and methylation specific PCR assay	19	Not specified	MGMT promoter methylation GBM patients serum and cerebrospinal fluid CSF samples for ctDNA detection using methylation specific PCR assay,	Detected 37% ctDNA in serum and 61% ctDNA in CSF. MGMT promoter methylation was detected with higher sensitivity in CSF (72.0%) compared to serum (41.7%), suggesting CSF as a promising tool for early diagnosis, treatment monitoring, and recurrence detection.	[66] (2015)
DNA extraction and nested PCR-based assays	38	Not specified	ctDNA was analysed for *TERT* promoter mutations (C228T and C250T) and *IDH* hotspot mutations.	Detected 8% (3 of 38 patients) ctDNA in plasma and 92% (35 of 38 patients) ctDNA in CSF.	[67] (2018)
DNA extraction and WGS	11	Not specified	Tumour-specific mutations in the ctDNA extracted from CSF.	100% ctDNA detection rate in CSF.	[68] (2015)
DNA extraction and WGS	13	Not specified	Copy number alterations in ctDNA.	50% ctDNA detection rate in CSF. Identified copy number alterations in the ctDNA, which closely reflected tumour genetic profile. Fragmentation patterns in CSF-derived ctDNA Copy number alterations in ctDNA were consistent with tumour tissue.	[79] (2018)
DNA extraction and NGS	16	Not specified	*IDH1/2* and *1p/19q* codeletion.	ctDNA detected in 49.4% of CSF samples. Genetic alterations detected closely matched those found in tumour biopsies.	[80] (2018)
DNA extraction and NGS	NA	Not specified	*IDH1* (R132H variant), *TERT* promoter (*C228T* mutation), *TP53*, *ATRX*, *H3F3A* and *HIST1H3B*.	CSF ctDNA more accurately reflected BM mutations, detecting all mutations in 83.33% of cases versus 27.78% for plasma ctDNA. CSF ctDNA more accurately reflected BM mutations, detecting all mutations in 83.33% of cases versus 27.78% for plasma ctDNA. Mutant allele frequency (MAF) in CSF ctDNA strongly correlated with BM tumour size (r = 0.95) and was higher than in plasma ctDNA (38.05% vs. 4.57%). MAF and tumour mutational burden in CSF ctDNA closely matched BM values (r = 0.96 and 0.97, respectively). CSF ctDNA exhibited superior concordance with BM (99.33%) compared to plasma ctDNA (67.44%), improving the identification of clinically relevant mutations. However, for multiple BM, plasma ctDNA performed well, achieving a 93.01% concordance, comparable to CSF ctDNA.	[81] (2023)
DNA extraction and MSK-IMPACT™, a NGS assay	711	Not specified	The distribution of clinically actionable somatic alterations was consistent with tumour-type specific alterations across the AACR GENIE cohort.	Genetic alterations were detected in 53% (489/922) of CSF samples from patients with confirmed CNS tumours, while none of the 85 samples from patients without CNS tumours contained detectable ctDNA. The identified mutations aligned with tumour-type-specific alterations observed in the AACR GENIE cohort. Repeated ctDNA testing revealed clonal evolution and resistance mechanisms, and the presence of ctDNA linked to reduced overall survival after CSF collection.	[82] (2024)
QIAamp Circulating Nucleic Acids kit (QIAGEN), Western blot, histopathology, and immunochemistry, digital PCR, and shallow whole genome sequencing were utilised	64	Not specified	Patient-derived orthotopically implanted xenograft models of GBM.	Fragment length profiling of host (rat) and tumour-derived (human) ctDNA revealed a 145 bp peak in the human fragments, suggesting differences in ctDNA origin or processing. ctDNA concentration was found to correlate with cell death, but only following temozolomide and radiotherapy treatment. Detection of tumour mitochondrial DNA (tmtDNA) in plasma using ddPCR, offering an alternative to nuclear ctDNA significantly increased detection rates (82% vs. 24%) and enabled tumour DNA detection in both CSF and urine. The plasma contained approximately 13 times more tmtDNA (558 copies) than CSF (43 copies), indicating that the BBB does not entirely restrict the release of tumour DNA.	[70] (2019)
QIAamp DNA micro kit (Qiagen) and amplicon sequencing	20	Not specified	*DH1, IDH2, TP53, TERT, ATRX, H3F3A*, and *HIST1H3B* gene mutations	Genomic analysis of ctDNA extracted from CSF enables the molecular subtyping of diffuse gliomas, aiding both surgical decision-making and clinical management.	[83] (2018)
QIAamp Circulating Nucleic Acid kit and NGS	26	Not specified	Cancer genomic panel sequencing on the CSF-derived ctDNA.	ctDNA was detected in the CSF of 24 out of 26 patients (92.3%). There was a high concordance between ctDNA and tumour DNA mutations, particularly for non-copy number variants and in GBM cases. Additionally, tumour mutational burden measured from CSF ctDNA strongly correlated with that of tumour tissue (R^2^ = 0.879, *p* < 0.001), with an even stronger correlation observed in GBM (R^2^ = 0.992, *p* < 0.001).	[84] (2022)
DNeasy Blood and Tissue Kit (Qiagen), and targeted DNA sequencing by Oxford Nanopore Technology MinION device.	12 paediatric high-grade glioma patients and 6 controls	0.1 femtomoles DNA (~0.3 pg)	Analysis of ctDNA with a handheld platform (Oxford Nanopore MinION) to quantify patient-specific CSF ctDNA variant allele fraction (VAF)	Nanopore sequencing achieved 85% sensitivity and 100% specificity in CSF samples (*n* = 127 replicates), with a detection limit of 0.1 femtomoles of DNA and a 12-h turnaround time, showing favourable comparison to NGS. Multiplexed analysis enabled simultaneous detection of *H3.3A* (*H3F3A*) and *H3C2* (*HIST1H3B*) mutations in a patient who had not undergone biopsy, with results validated by ddPCR. Serial ctDNA sequencing from CSF using Nanopore correlated with radiological response in a clinical trial, including one case where a strong multi-gene molecular response predicted durable clinical benefit.	[85] (2020)

### 2.3. Nucleosomes

Chromatin’s structure was first described in 1974 as repeating nucleosome units. Each nucleosome consists of an octamer of two copies of each of the core histones H2A, H2B, H3 and H4, wrapped by 146–147 base pairs of DNA [86,87]. Nucleosomes are generated by DNA associated with proteins, including histone H1. Histones have a globular domain for histone–histone interactions and flexible, positively charged tails (20–35 amino acids) that undergo post-translational modifications (PTMs) [88]. As the basic units of chromatin, nucleosomes regulate key nuclear processes like transcription, replication and repair, primarily through PTMs [89]. These modifications influence gene expression on an additional level to DNA methylation, and activating oncogenes or silencing tumour suppressor genes may contribute to cancer development and progression [89,90]. Histone PTMs include methylation, acetylation, phosphorylation, ubiquitination, sumoylation, glycosylation, homocysteinylation, and crotonylation [91]. Methylation may activate or repress gene expression depending on the site and degree [89], while acetylation generally promotes transcription by neutralising lysine’s positive charge, reducing chromatin compaction and enhancing accessibility for transcription factors [91]. The interplay of these PTMs forms a “histone code” read by cellular proteins to control transcription, replication, and repair, with core histone variants adding further complexity [91,92].

Nucleosomes and their PTMs can be detected in plasma or serum using immunoassays such as the chemiluminescence immunoassay (ChLIA) or enzyme-linked immunosorbent assay (ELISA) (Table 3). These assays detect specific biomarkers from complex matrices, and can be applied on automated platforms which would allow for faster and more reproducible results [93]. Cancer cell death, including that of GBM cells, results in the release of nucleosomes into the bloodstream, which are transported as mono- or oligo- nucleosomes bound to ctDNA. 

Cancer patients have a greater count of circulating nucleosomes due to an increase in cellular turnover in comparison to healthy controls and due to the cytotoxic effect of treatments which lead to cell death [89]. High levels of nucleosomes are not specific to cancer or GBM per se, since it is also linked to trauma, stroke and sepsis, which limits its clinical use as a unique biomarker for GBM detection. The latest studies have, however, reported that the PTMs found in circulating nucleosomes may have cancer specificity and are thus be investigated as biomarkers [94]. 

Enzymes that regulate PTMs are often dysregulated in GBM. Histone deacetylases and demethylases like lysine-specific histone demethylase 1, can alter the epigenetic status of brain cells, influencing cancer development and progression [94,95]. In paediatric high-grade gliomas, recurrent histone mutations are common. The *H3K27M* mutation, a lysine-to-methionine substitution at position 27 of H3.1 or H3.3 histone genes, is associated with childhood diffuse intrinsic pontine glioma, disrupting polycomb repressive complex 2 function, leading to global *H3K27* methylation loss and altered gene expression [96]. *H3G34R/V* mutations, involving glycine 34 substitutions in H3.3 are seen in gliomas of the cerebral hemispheres in both children and adults causing redistribution of the activating H3K36 methylation mark and transcriptional dysregulation [94,96]. While this review focussed on adult glioma, the analysis of liquid biopsy in paediatric glioma with often different biology faces other problems such as sample volume, which has recently been in detail reviewed by others [97,98,99,100].

Another epigenetic marker dysregulated in GBM is histone 3 lysine 4 (H3K4), which is decreased in aggressive GBM, leading to gene repression [95]. Acetylation of histone 3 at lysine 18 (H3K18Ac), is another common PTM with dysregulated expression seen in a number of cancers including GBM, where lower levels imply a better prognosis for the patient [101]. Accordingly, a profile of PTMs such as circulating nucleosome-linked histone modifications has the potential to be studied in GBM patients that could be added to MRI, as a guide to diagnosing and monitoring tumour progression in GBM patients [9].

Nucleosomes are generally less useful for cancer diagnosis due to their elevated levels in benign diseases. However, in gastrointestinal cancers, nucleosome levels correlate with tumour stage and metastasis. Nucleosomes are more valuable for therapy monitoring, as their levels decrease in remission and increase with disease progression during chemotherapy and radiotherapy, a pattern observed in lung, pancreatic, colorectal cancers, and haematological malignancies [102,103]. In GBM, circulating nucleosomes in serum and CSF have shown potential for diagnosis and prognosis. Patients developing cerebral oedema post-surgery exhibited a ~200-fold rise in CSF nucleosomes. Monitoring these levels may help detect complications and track tumour progression and treatment response [103]. However, further research is needed to refine their specificity and clinical application, especially in distinguishing cancer from other conditions.

Collectively, circulating nucleosomes and their PTMs present a promising avenue for GBM detection and monitoring. While total nucleosome levels in plasma and CSF lack disease specificity, unique histone modifications, such as H3K27M and H3K18Ac, provide greater diagnostic potential. Emerging technologies, including immunoassays and single-molecule detection methods, have enabled precise profiling of nucleosome-bound epigenetic markers, distinguishing GBM patients from healthy individuals. These findings suggest that nucleosome-based liquid biopsies could serve as a valuable adjunct to MRI and conventional diagnostics, particularly in monitoring disease status and therapy response. Additionally, nucleosome levels may help identify postoperative complications, such as cerebral oedema, offering further insight into tumour progression and treatment outcomes. However, further validation in larger clinical studies is necessary to refine their specificity and clinical application, especially in differentiating cancer from other conditions.

**Table 3 cancers-17-02700-t003:** Nucleosome isolation and analysis in patients with GBM.

Isolation Method	Patient Numbers	Markers Used to Verify GBM Origin	Findings	Reference
Cell Death Detection ELISA Plus kit	10	NA	Pre-therapeutic nucleosome levels in both serum and CSF did not significantly differ among GBM patients and control groups. Postoperative Increase: In GBM patients, nucleosome levels in serum and CSF increased moderately during the week following surgery and intracavitary chemotherapy. Cerebral Oedema Correlation: Three out of ten GBM patients developed cerebral oedema post-surgery. In these patients, CSF nucleosome levels increased almost 200-fold, peaking on day 3 postoperatively. In contrast, the seven patients without oedema exhibited only slight increases in nucleosome levels. Clinical Implication: Monitoring CSF nucleosome levels may serve as an indicator for postoperative complications such as cerebral oedema in GBM patients.	[103]
Single-molecule technology to detect and monitor plasma-circulating nucleosomes	NA	*H3K27M* mutation and mutant *p53*	The single-molecule analysis revealed epigenetic patterns unique to diffuse midline glioma, enabling differentiation from healthy individuals and patients with other cancer types. This approach profiles multiple histone modifications on individual nucleosomes from less than 1 mL of plasma, revealing epigenetic patterns unique to glioma that significantly differentiate these patients from healthy individuals and those with other cancer types. The detection strategy demonstrated a correlation with MRI measurements and ddPCR assessments of ctDNA, highlighting its potential utility in non-invasive treatment monitoring. Suitable for paediatric patients and scenarios where sample volume is limited.	[104]

### 2.4. Circulating Tumour Microrna

Circulating tumour microRNA (miRNA), is the most common small non-coding RNA, typically 21–23 nucleotides long, and regulates up to 30% of protein-coding genes [105,106]. It plays a key role in cancer by downregulating tumour suppressor genes. miRNAs are highly stable and detectable in bodily fluids such as blood, urine, and CSF, and are also present in extracellular vesicles (EVs) [12,107,108,109]. 

Altered expression of specific miRNAs have been found to distinguish between GBM patients and lower-grade glioma patients (Table 4). For instance, downregulation of miR-125b, miR-16, miR-497, miR-128, miR-342-3p, miR-205 and upregulation of miR-210, miR-182, miR-20a-5p, miR-454-3p, miR-106a-5p, miR-181b-5p have been linked to GBM pathophysiology [110,111,112,113,114,115,116,117]. Thus, miRNA dysregulation in GBM involves both upregulation of oncogenic miRNAs (“oncomirs”) and downregulation of tumour-suppressive miRNAs, each playing distinct roles in tumour biology. 

Most miRNAs which show an increase in their expression level in GBM are oncogenic and are known as “oncomirs” [118]. Oncomirs usually promote tumour growth by inhibiting tumour suppressor genes and genes linked to cell differentiation or apoptosis. The first oncomir which showed significantly increased levels in GBM tissue samples compared to normal tissue was miR21. It can target several tumour suppressors such as PTEN [119]. The levels of miR-21 in GBM tissue and plasma samples was higher in grade II and III GBM samples compared to controls, yet there was no significant change between pre-surgery and post-surgery sample measures, while the miR21 level was decreased after chemo irradiation therapy [110].

Several miRNAs function as tumour suppressors and are commonly downregulated in GBM patients [120]. Tumour-suppressive miRNA such as Let-7, which targets oncogenes like *K-RAS* and *MYC* inhibit GBM cell proliferation by interfering with histone methyltransferase *EZH2* [121,122,123]. miR-128 and miR-342-3p, significantly reduced in the plasma of GBM patients, have been shown to return to normal levels after surgery and chemo-radiation, indicating the patient’s response to treatment [111]. These miRNAs are glioma-specific, and their loss or downregulation may contribute to tumour progression including the malignant transformation of meningioma and pituitary adenomas into GBM [124,125]. A meta-analysis demonstrated that serum miR-125b, markedly reduced in GBM patients, shows a 3.5-fold higher detection frequency compared to healthy controls, highlighting its potential as a screening biomarker [126]. Similarly, miR-205, another tumour suppressor, was significantly decreased in GBM patients, increased post-surgery, and declined again upon recurrence, suggesting its value as a dynamic biomarker [116]. Accordingly, the dysregulated expression of miRNA, shapes the prediction of cancer, early diagnosis, prognosis and histological classification [9,107]

Other studies have suggested that as the BBB is not permeable for some miRNAs, it may be more valuable to profile miRNA from CSF samples. As miR-10b is greatly overexpressed in GBM but absent in normal brain, it can be detected in CSF of majority of GBM patients (89%) but not in the CSF of healthy controls. Considering extracranial tissues express miR-10b, its absence in the CSF of healthy controls suggests that it may not leak through the BBB under non-neoplastic conditions [107,127]. 

Several studies have identified specific miRNAs that are dysregulated in GBM patients, with both upregulated and downregulated miRNAs linked to tumour progression and prognosis. miRNAs such as miR-21, miR-128, and miR-342-3p have been highlighted for their potential as biomarkers for early GBM detection, offering high sensitivity and specificity, particularly in blood and urinary samples. These clinical data signify that the dysregulation of miRNAs, whether through upregulation or downregulation, can be used to diagnose or monitor GBM patients. Though the data are thought provoking, caution should prevail due to the small size of studies, and the lack of a standard method for blood collection, RNA extraction and sequencing. Of note is that specificity of miRNA detection may be lower than that of ctDNA [9], but it may be considered as complementary to ctDNA detection. Despite challenges, miRNA-based liquid biopsy offers a promising, less-invasive alternative to traditional tissue biopsies for GBM diagnosis, prognosis, and treatment monitoring.

**Table 4 cancers-17-02700-t004:** miRNA isolation and analysis in patients with GBM.

Isolation Method	Patient Numbers	Markers Used to Verify GBM Origin	Findings	Reference And Publication Date
miRNA extraction and qPCR	20	miR-221 and miR-222	Both miR-221 and miR-222 were significantly upregulated in the plasma of GBM patients compared to healthy controls. miR-221 demonstrated 90% sensitivity and 100% specificity and miR-222 demonstrated 85% sensitivity and 100% specificity for GBM detection. Expression levels of miR-221 and miR-222 decreased following treatment.	[128] (2019)
miRNA profiling performed using the Nanostring^®^ (Seattle, WA, USA) platform	91	miR-223 and miR-320e, *IDH* mutation status and *1p/19q* co-deletion.	Dynamic changes in miR-320e were linked to tumour volume in GBM patients. A 9-miRNA signature was established, distinguishing glioma patients from healthy controls with 99.8% accuracy. miRNA levels did not increase in cases of pseudo-progression. This supports their use in distinguishing true progression from treatment effects and in post-operative monitoring	[129] (2020)
mirVana™ miRNA Isolation Kit and qRT-PCR	50	miR-21, miR-128, and miR-342-3p	miR-21 was significantly upregulated, while miR-128 and miR-342-3p were markedly downregulated in glioma patients compared to healthy controls. Notably, miR-21 demonstrated a high diagnostic performance with 90% sensitivity, and 100% specificity.	[110] (2012)
RNA isolation and NGS, including mRNA-seq and small RNA-seq	7	mRNA and miRNA candidates	The study identified differentially expressed genes in individual patients, with up to 93 mRNA and 19 miRNA candidates linked to GBM recurrence.	[130] (2024)
Urinary microRNA-based diagnostic model for CNS tumours using nanowire scaffolds.	119	Differential miRNA expression profiles	The study reported high diagnostic performance, with sensitivity and specificity values of 100% and 97%, respectively, for detecting early-stage CNS tumours. Non-invasive method holds promise for early detection and monitoring of CNS tumours through urine-based liquid biopsy.	[131] (2021)
miRCURY RNA Isolation Kit and qRT-PCR	10	miR-21, miR-218, miR-193b, miR- 331, and miR-374a, miR- 548c, miR-520f, miR-27b, and miR-130b	Sampling of CSF from the lumbar region to extract 9 signature miRNA for GBM. The overexpressed signatures were miR-21, miR-218, miR-193b, miR-331, and miR-374a, while the down regulated were miR-548c, miR-520f, miR-27b, and miR-130b in GBM CSF. The study compared the diagnostic performance of miRNA detection between CSF obtained from the cisternal and lumbar regions. The cisternal CSF samples demonstrated a sensitivity of 80% and specificity of 67% for GBM detection, whereas the lumbar CSF samples showed a sensitivity of 28% and specificity of 95%.	[132] (2017)
TaqMan Low Density Array platform for miRNA profiling and qRT-PCR	16 GBM and 9 healthy patients	miR-451, miR-711, miR-935, miR-223	Showed that miRNA from CSF can differentiate between tumour and non-tumour diseases states. Identified distinct miRNA signature in CSF that can distinguish CNS malignancies (GBM) from non-tumour controls.	[133] (2015)

### 2.5. Extracellular Vesicles 

Extracellular vesicles (EVs), including exosomes and microvesicles (MVs) are small membrane-enclosed particles released by normal and tumour cells. They serve as carriers of macromolecules in liquid biopsies of brain tumour patients [18]. EVs are heterogeneous in size, quantity and origin of molecular content and biological activity. Two main groups of EVs exist, which differ in origin and size. MVs (50 to 500 nm) originate from the cell membrane, while exosomes (50 to 150 nm) originate from the endosomal system [134,135]. EVs protect their cargo, mRNAs, miRNAs, lipids or proteins, which are specific to the cell origin [136], from enzymatic degradation via their lipid bilayer, enabling them to cross the BBB [134]. 

The significance of EVs is emphasised by the fact that their transcriptomic and proteomic profile is specific to the cell of origin and can differ in response to diverse stimuli [10]. Their imperative role in intercellular communications allows them to alter the phenotype of recipient cells by transferring genetic information and proteins [12,137]. These molecules often reflect malignant processes, underlining their importance as liquid biopsy tools in cancer, including GBM [136]. EV-mediated crosstalk has been demonstrated between GBM and its tumour microenvironment, fostering tumour progression [138]. Exosomes from hypoxic GBM cells overexpress *VEGF-A*, disrupting the BBB by downregulating occludin and claudin-5 [139]. 

EV isolation typically involves immunoaffinity capture or differential centrifugation gradients [9]. Exosomes are characterised by transmission electron microscopy (TEM), nanoparticle-tracking analysis (NTA) and surface markers such as CD63, CD81, CD9, CD37, CD53, CD82, ICAM-1 and integrins, detectable by flow cytometry or Western blot [136].

Exosome-based liquid biopsies show promise in cancer diagnostics, with tests like ExoDx™ Lung (ALK) and ExoDx Prostate IntelliScore (EPI) already approved. For GBM, research is ongoing [140]. Emerging biomarkers include exosomal circular RNAs (circRNAs), EGFRvIII RNA for monitoring CAR-T cell therapy response, and MGMT methylation for predicting treatment outcomes. Despite promising findings (see Table 5), no exosome-based GBM test has yet reached clinical practice, pending further validation in larger cohorts [140,141,142,143,144].

Studies indicate that EVs from GBM stem cells exhibit increased adhesion-related proteins after TMZ treatment, potentially promoting tumour progression [145]. TMZ resistance may be associated with EVs carrying MGMT mRNA, as a potential marker of resistance, alongside adhesion proteins (e.g., TGM2, CD44 and CD133), and stemness markers like NESTIN [146,147,148]. These EV-associated molecules may serve as biomarkers to monitor drug resistance and treatment failure [10]. 

Overall, EVs, particularly exosomes, are promising biomarkers for GBM detection, monitoring, and prognosis. Exosomal RNA, proteins, and lipids include disease-specific molecular cargo, such as EGFRvIII RNA and syndecan-1 with high diagnostic potential. EV-derived molecular cargo may reflect or potentially even contribute to treatment resistance and tumour progression, though their mechanistic role remains under investigation. Although exosome-based liquid biopsy presents a minimally invasive alternative to traditional tissue biopsies, larger patient cohorts need to be analysed and standardised isolation and characterisation techniques are needed to validate their utility in GBM. Despite these limitations, ongoing research suggests that EV-based liquid biopsies could revolutionise GBM management, offering real-time insights into disease progression and therapeutic response.

**Table 5 cancers-17-02700-t005:** EV isolation and analysis in patients with GBM.

Isolation Method	Patient Numbers	Markers Used to Verify GBM Origin	Findings	References and Publication Date
Differential centrifugation and flow cytometry	11	GFAP	Concentration and composition of circulating MVs in patient plasma correlated with tumour progression. Elevated levels of tumour-derived MVs were associated with true tumour progression, while lower levels were indicative of treatment-related changes or pseudoprogression.	[149] (2014)
Differential centrifugation and ultracentrifugation, filtration and flotation density gradient centrifugation.	25	Tumour-specific EGFRvIII mRNA within the vesicles	EVs contain functional RNA and proteins that may influence the tumour microenvironment and serve as diagnostic tools GBM cells release MVs containing mRNA, miRNA, and angiogenic proteins. Detection of tumour-specific EGFRvIII mRNA in serum-derived MVs supports their potential as non-invasive biomarkers for GBM diagnosis and monitoring	[150] (2008)
Serial centrifugation and flow cytometry	16	Annexin V, CD41, CD235 and Anti-EGFR	Rising levels of Annexin V-positive MVs during chemoradiation therapy correlated with earlier tumour recurrence and reduced overall survival. Patients with higher levels of MVs had >4-fold increase in the hazard ratio for recurrence compared to those with lower levels The study provided initial evidence that monitoring blood-borne MV levels could serve as a non-invasive method to predict disease progression and patient outcomes in newly diagnosed GBM patients.	[151] (2016)
Ultrafiltration and ultracentrifugation and NTA	43	GFAP	Plasma EV concentrations were significantly elevated in GBM patients compared to healthy controls (*p* = 0.0099). The average EV size was comparable between GBM and healthy groups in both the discovery (*p* = 0.548) and validation cohorts (*p* = 0.075). Circulating EV levels showed no correlation with tumour size (*p* = 0.318). However, the degree of necrosis significantly impacted EV secretion (*p* = 0.045), with higher necrosis (grade 3) in GBM samples markedly reducing EV release. Elevated EV levels in GBM plasma decreased post-surgery and rose at recurrence.	[152] (2019)
Precipitation using ThermoFisher kit and Semi quantitative RT-PCR	96	EGFRvIII mRNA within the vesicles	EGFRvIII prevalence in the dataset was 39.58%. The sensitivity and specificity of serum EV analysis for EGFRvIII was 81.58% (95% CI 65.67–92.96%) and 79.31% (95% CI 66.65–88.83%), respectively	[153] (2018)
Differential centrifugation and qRT-PCR	60	miR-301a	Serum exosomal miR-301a levels were significantly elevated in glioma patients compared to healthy controls. Higher miR-301a levels were associated with higher tumour grades and lower Karnofsky Performance Status (KPS) scores. Post-surgical samples showed a significant reduction in miR-301a levels, which increased again during tumour recurrence, suggesting its potential as a marker for disease monitoring. Kaplan–Meier survival analysis indicated that patients with higher serum exosomal miR-301a levels had shorter overall survival.	[154] (2018)
Ultracentrifugation and TEM	42	miR-320, miR-574-3p and RNU6-1	RNU6-1 identified in serum exosomes could effectively distinguish GBM patients from healthy individuals. The elevated levels of RNU6-1 in GBM patients’ exosomes suggest its potential as a non-invasive diagnostic biomarker.	[155] (2014)
Differential centrifugation and qPCR	12	miR-182-5p, miR-328-3p, miR-339-5p, miR-340-5p, miR-485-3p, miR-486-5p and miR-543	The identified miRNA signature in serum exosomes could effectively distinguish GBM patients from healthy individuals and those with lower-grade gliomas.	[108] (2018)
Density gradient ultracentrifugation, using OptiPrep™ Density Gradient Medium and sequencing followed by differential expression analysis	12 GBM (astrocytoma grade IV) and 5 astrocytoma grade II–III	Analysis of CUSA (cavitron ultrasonic surgical aspirate) EV and serum EV miRNA and piRNA	Seven miRNA species (miR-182-5p, 382-3p, 339-5p, 340-5p, 485-3p, 486-3p, and 543) were identified as the most reliable classifiers for GBM, achieving an overall predictive accuracy of 91.7%. Moreover, multivariate models using six different combinations of these markers were able to distinguish GBM patients from healthy controls with perfect accuracy (100%).	[109] (2020)
Chemical precipitation using ExoQuick-TC and qPCR	100 patients with glioma, 11 with metastatic brain tumours, 30 healthy patients	Expression of 3 miRNAs: miR-21, miR-222 and miR-124-3p, in serum exosomes.	Exosomal miR-21, miR-222, and miR-124-3p demonstrated strong discriminatory power in distinguishing GBM patients from healthy individuals, with AUC values of 0.84 (95% CI: 0.7538–0.9371, *p* < 0.001), 0.80 (95% CI: 0.6967–0.8980, *p* < 0.001), and 0.78 (95% CI: 0.6732–0.8904, *p* < 0.001), respectively. Among these, miR-21 showed the highest diagnostic accuracy for differentiating high-grade glioma from low-grade glioma, achieving an AUC of 0.83 (95% CI: 0.7395–0.9398, *p* < 0.001).	[156] (2018)
Ultracentrifugation, NTA and TEM for plEV isolation with proximity extension assay–based ultrasensitive immunoprofiling.	82	Syndecan-1	SDC1 in plEVs could discriminate between GBM and low-grade glioma with a sensitivity of 71%, and specificity of 91%. The findings support the concept of circulating plEVs as a tool for non-invasive diagnosis and monitoring of gliomas.	[157] (2019)
Total Exosome Isolation reagent followed by ultracentrifugation and qRT-PCR	43 GBM, 23 other brain tumour patients and 40 healthy individuals	Serum analysed for presence of EVs and HOTAIR biomarker	HOTAIR can be used as a biomarker for Dx and progression of some brain tumours including GBM with sensitivity and specificity of HOTAIR 86.1% and 87.5%, respectively.	[158] (2018)
Serial ultracentrifugation and short non-coding RNA sequencing using the OASIS-2.0 platform.	5	Isolate EV’s from human differentiated GBM cells in vitro and perform short, non-coding RNA sequencing to determine expression pattern	Small genome sequencing identified 712 non-coding RNA sequences, the majority of which had not been previously linked to GBM-derived EVs. These included members of the let-7 miRNA family, miR-3182, miR-4448, miR-100-5p, and miR-27-3p. In addition, several non-miRNA short non-coding RNA types were detected, such as piRNA, snRNA, snoRNA, and yRNA.	[159] (2020)
qPCR	5 Glioma patients before and after radiotherapy	Expression signature of miRNAs in glioma patients before and after radiotherapy	Eighteen upregulated and sixteen downregulated differentially expressed (DE) miRNAs were identified, and their target genes were predicted using multiple miRNA–target interaction databases.	[160] (2020)
Differential centrifugation and DNA analysis through methylation array analysis Proteome analysis with differential quantitative proteomics	Unspecified number of glioma patients and non-tumorous temporal tissue from patients undergoing epilepsy surgery	DNA and protein analysis of Glioma and non-tumorous EVs	Tumour-specific mutations and copy number alterations were identified in EV-DNA with high accuracy. However, proteomic analysis was insufficient for accurate tumour classification or identification.	[161] (2021)
Size-exclusion chromatography (by using qEV columns from IZON^®^ (IZON CO., LTD., Seongnam, Republic of Korea)), followed by immunoprecipitation with CD44-conjugated beads and qRT-PCR	55	Suitability of novel EV isolation procedure and analysis of serum EVs for miRNA biomarkers and their correlation with prognosis.	Four serum biomarkers were identified as predictive of prognosis, miR-15b-3p, miR-21-3p, and miR-328-3p were negatively correlated with prognosis, with elevated expression levels indicative of poorer outcomes, whereas miR-106a-5p showed a positive correlation, with higher levels linked to improved prognosis prediction of GBM.	[162] (2020)
Differential centrifugation and NTA	96 total patients, 24 GBM, 24 meningioma, 24 BM from NSCLC and 24 controls from patients with benign disc herniation.	Proteomic analysis of serum and serum derived small EVs	A 10-protein panel for whole serum and a 17-protein panel for sEV samples were identified. Although no single protein could independently differentiate between patient groups, the combined panel effectively distinguished among them. This indicates that accurate classification of tumour types may depend on a specific set of proteins rather than individual biomarkers	[163] (2020)
Ultracentrifugation and RNA extraction from EVs, ddPCR and flow cytometry for rare mutation detection.	14 glioma patients (4 with GBM)	Mutant *IDH1* G395A	CSF is a viable bio fluid to examine contents of EVs. Mutant *IDH1* G395A identified in CSF EVs with a sensitivity of 63% and specificity of 100%. CSF EVs generally contained higher levels of mutant mRNA than serum EVs. EVs carry tumour-specific RNA signatures that can be detected non-invasively, supporting the potential of EVs as liquid biopsy tools for glioma diagnosis and monitoring.	[164] (2013)
Ultracentrifugation and purification with sucrose cushion for EV isolation. qRT-PCR for mRNA transcript detection.	25 GBM, 5 low-grade glioma and 4 healthy patients	EGFRvIII mutant mRNA	EGFRvIII oncogene, in EV RNA can be accomplished with a sensitivity of 60% and 98% specificity in comparison to the gold standard qPCR of EGFRvIII transcript from brain tumour tissue.	[165] (2017)
Ultracentrifugation and. TEM for EV isolation. Western blotting for exosomal markers. qRT-PCR for mRNA transcript quantification.	9 GBM and 5 healthy patients	Exosomal markers (CD63 and TSG101) miR-21	Showed that miR-21 from CSF EVs can differentiate between tumour and non-tumour diseases states.	[148] (2013)
Ultracentrifugation and OptiPrep™ density gradient ultracentrifugation for further purification. NTA, flow cytometry and Western blotting (to detect PD-L1 on EV surface).	10	Patient derived GBM stem cells.	*PD-L1* expression on the surface of GBM derived EVs, can prevent T-cell activation and proliferation upon binding directly to *PD-L1*. This indicates *PD-L1* expression on EVs can be an immune-escape mechanism for GBM	[166] (2018)
Differential centrifugation of MVs and microfluidic chip-based immunomagnetic technique called μNMR (miniaturised nuclear magnetic resonance) for protein typing of EVs/MVs. Also used magnetic nanoparticles (MNPs) conjugated to antibodies to detect specific tumour-associated proteins on circulating MVs.	15	EGFRvIII mutant, PDPN and IDH1	Tumour-derived MVs carrying EGFR/EGFRvIII proteins were successfully detected in patient plasma. Showed four protein panels of EV surface proteins can be used to discriminate GBM patients from healthy controls using novel antibody capture method. The system enabled real-time monitoring of tumour progression and treatment response by tracking changes in circulating tumour-derived MV profiles.	[167] (2012)
Proteomics using mass spectrometry	22	MGMT and IDH statuses and GAP43	CSF proteomics using mass spectrometry enables GBM biomarker discovery from small volumes (~30 μL). Mikolajewics et al., identified 755 unique proteins in 73 CSF samples (22 GBM), with MGMT and IDH statuses accurately detected at 94.1% and 33.3%, respectively. Single-cell RNA sequencing confirmed GAP43 as GBM-specific, while TFF3 and CACNA2D2 were specific to BM and CNS lymphoma.	[168] (2022)
Proteomics with sequential window acquisition of all theoretical fragment ion spectra mass spectrometry (SWATH-MS) and immunohistochemistry for validation.	134	*BCAS1*, *INF1*, and *FBXO2*	Identified overexpressed proteins CSF proteomics in recurrent GBM, to quantify the proteomes of newly diagnosed and recurrent GBM patients and validated the markers using immunohistochemistry.	[169] (2023)
Lipidomics using Quadrupole time-of-flight liquid mass spectrometer Q-TOF LCMS/MS	14 GBM and 14 healthy patients	NA; based on statistically significant differences in blood lipid species between GBM and controls	Lipidomics also holds potential. Identified differential lipid species including fatty acids, saccharolipid, sphingolipid, glycerolipid and sterol lipid from blood samples.	[170] (2022)

## 3. Application of Machine Learning and Artificial Intelligence 

The diagnosis of glioblastoma using signals derived from liquid biopsy samples remains particularly challenging, traditional statistical approaches frequently fall short in detecting the nonlinear patterns and subtle molecular variations that characterise GBM biology, increasing the risk of misdiagnosis or missed detection when relying solely on liquid biopsy data. To address these challenges, artificial intelligence (AI) and machine learning (ML) approaches are increasingly integrated into GBM liquid biopsy workflows. Novel ML algorithms can extract meaningful insights from noisy and heterogeneous datasets, enabling earlier and more accurate detection of GBM tumours, even when biomarker levels are extremely low in blood or CSF. Various liquid biopsy-based signatures have shown potential for tumour detection, including plasma denaturation profiles, CSF proteomic signatures, and serum miRNA signatures [129,168]. In omics analysis for GBM, ML techniques uncover complex molecular signatures linked to tumour subtype classification, disease progression, and therapeutic response [171,172]. This allows for the development of robust predictive models that outperform traditional biomarker-based approaches, advancing precision diagnostics and personalised treatment strategies for glioblastoma patients. Once trained, these models can be applied in clinical practice [171,172]. 

Fragmentation patterns and personalised cfDNA sequencing in urine and plasma have differentiated glioma patients from healthy individuals [173]. Tumour-educated platelets (TEPs), analysed with swarm algorithms, have identified spliced RNA biomarkers to distinguish false positives from true disease progression [174]. 

ML has also identified tumour-specific DNA methylation markers in blood and tumour-associated MRI features [175,176]. In several studies, ML algorithms identified MRI based tumour-associated features, while serum spectroscopy combined with MRI detected spectra variations between healthy and tumour-affected patients. Additionally, ML has been applied to investigate the association between tumour size and the effectiveness of liquid biopsy testing. In a study involving 177 patients including 90 with high-grade gliomas (such as GBM or anaplastic astrocytoma) and others with low-grade gliomas (astrocytoma, oligoastrocytoma, and oligodendroglioma), a spectroscopic liquid biopsy method successfully identified small and low-grade tumours, supporting early diagnosis [177,178].

AI enhances liquid biopsy applications and diagnostic performance by enabling rapid analysis of cancer-related circulating proteins and nucleic acids and enhancing imaging from subjective interpretation to quantitative analysis [179,180,181]. Multimodal AI models integrate proteomic and genomic data from liquid biopsies and quantitative imaging to create a more comprehensive diagnostic pathway [182,183]. This helps distinguish benign from malignant lesions, reduces unnecessary follow-ups and guides therapeutic decisions.

Despite its potential, the application of AI and ML in liquid biopsy and cancer diagnostics faces several challenges. One major hurdle is the need for large, high-quality datasets to train robust models, as variability in sample collection, processing, and sequencing methods introduces bias [184]. Importantly, while integrating findings across biofluids provides a broad perspective, future studies should also recognise the distinct analytical considerations of each matrix. For example, CSF offers a higher CNS-specific biomarker abundance, reduced systemic noise, and unique analyte profiles, but requires low-input assay adaptations and safeguards against blood contamination. Incorporating such matrix-specific knowledge into AI/ML-driven multi-biofluid frameworks may accelerate clinical translation [185,186]. Additionally, the interpretability of AI-driven models remains a concern, as complex algorithms often function as "black boxes," making it difficult for clinicians to understand the reasoning behind predictions [172,182]. Regulatory approval and clinical validation also pose significant barriers, requiring extensive trials to ensure reliability and accuracy. Finally, integrating AI into existing healthcare systems demands substantial computational resources, infrastructure, and clinician training, which may limit widespread adoption [176,182,184,187,188].

## 4. Conclusions

In the past, clinicians may have ignored the liquid biopsy literature, due to the difficulties in reliable detection of rare events, but as more data is presented, and as techniques and methodologies improve, liquid biopsies, particularly for patients with brain cancer (primary and metastatic), offer a compelling advance in brain cancer therapy. Increasingly, where tissue access is contra-indicated (e.g., brain stem gliomas, tumours in eloquent areas), not recommended (e.g., patients with co-morbidities or frailty), or inconvenient (e.g., CSF monitoring), liquid biopsies, alone or in conjunction with traditional imaging, may represent the best way to aid in the management of these patients. Although this review cannot always confirm the use of the latest WHO classification, its findings remain relevant. In the coming years, clinical studies are expected to clarify which liquid biopsy techniques provide the most value for diagnosing, prognosing, and treating primary brain tumours, potentially surpassing traditional imaging and tissue-based methods. GBM liquid biopsy research is rapidly gaining traction. Analysis of body fluids, such as blood, or CSF, from GBM patients reveal various tumour-derived components, including CTCs, ctDNA, miRNAs, EVs, nucleosomes, and metabolites. These biomarkers may serve as alternatives to tissue samples and support genotype-guided therapies and personalised GBM management. They also show promise for monitoring tumour progression, treatment response, and guiding therapeutic choices. Imaging alone remains unreliable for distinguishing true progression from treatment effects, highlighting the need for adjunct approaches. Liquid biopsy markers vary in their technical demands, from easily measurable analytes to those requiring specialised platforms, and assessing their clinical utility requires more than sensitivity or specificity alone. Recent advances such as genome-wide fragmentomic analysis using machine learning [189] demonstrate the potential of scalable and innovative approaches in this space. However, accessing tumour-derived material is challenging due to the BBB, which limits the release of biomarkers into the bloodstream—currently the most studied liquid biopsy. Consequently, tumour-derived nucleic acids in serum or plasma often occur at low levels, hindering routine clinical use. Improving sensitivity will require optimising sample volumes, technological platforms, and the use of artificial intelligence. Combining the measurement of multiple different liquid biopsy-based markers from the same biopsy will likely improve overall sensitivity of single modality testing but this idea needs to be further validated in the future.

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
