# Peer review of "The Quest for Non-Invasive Diagnosis: A Review of Liquid Biopsy in Glioblastoma"

_cancers, 2025, doi:10.3390/cancers17162700_

Round 1

Reviewer 1 Report

Comments and Suggestions for Authors

This a thorough and comprehensive overview of efforts to identify non-invasive and minimally invasive biomarkers that may identify/grade brain tumors and track response.

A few suggestions

I believe on overview paragraph in the introduction or perhaps conclusion would be helpful that (1)more clearly defines the importance of the problem that imaging is not reliable at tracking outcome or even confirming diagnosis for readers who are not clinical GBM specialists (2) provides a preview of your objective of defining circulating factors that may be identified in either blood or the less pleasant acquisition of CSF (3) and that it includes markers that may or may not require highly skilled teams to measure.  This could provide a construct for the reader in assessing the potential general value of further exploration of each approach beyond the sensitivity and specificity identified  in existing studies that are generally small and/or without validation on independent data sets,

A recent reference of a novel approach including machine learning that may be discussed (i realize this is rapidly moving field):  Mathios D, Niknafs N, Annapragada AV, Bobeff EJ, Chiao EJ, Boyapati K, Boyapati K, Short S, Bartolomucci AL, Cristiano S, Koul S, Vulpescu NA, Ferreira L, Medina JE, Bruhm DC, Adleff V, Podstawka M, StanisÅ‚awska P, Park CK, Huang J, Gallia GL, Brem H, Mukherjee D, Caplan JM, Weingart J, Jackson CM, Lim M, Phallen J, Scharpf RB, Velculescu VE. Detection of brain cancer using genome-wide cell-free DNA fragmentomes. Cancer Discov. 2025 Apr 29. doi: 10.1158/2159-8290.CD-25-0074. Epub ahead of print. PMID: 40299319.

One sentence jumped out as confusing--

Liquid biopsies comprise the detection and quantification of tumour related content in bio- 69
fluids that is released by tumours, such as circulating tumour cells (CTCs), circulating 70
tumour DNA (ctDNA), circulating tumour microRNA (miRNA), extracellular vesicles 71
(EVs) and nucleosomes or in response to tumours, all of which can provide critical insights 72
into GBM progression and recurrence.

might be changed to something similar to:

Liquid biopsies comprise the detection and quantification of tumour related content in bio-
fluids that is released by tumours or in response to the presence of tumor, such as circulating tumour cells (CTCs), circulating tumour DNA (ctDNA), circulating tumour microRNA (miRNA), extracellular vesicles(EVs) and nucleosomes or in response to tumours, all of which can provide critical insights into GBM progression and recurrence.

Author Response

Reviewer 1

This a thorough and comprehensive overview of efforts to identify non-invasive and minimally invasive biomarkers that may identify/grade brain tumors and track response.

We thank the reviewer for recognizing the thoroughness and comprehensiveness of our review.

A few suggestions

  • I believe on overview paragraph in the introduction or perhaps conclusion would be helpful that (1) more clearly defines the importance of the problem that imaging is not reliable at tracking outcome or even confirming diagnosis for readers who are not clinical GBM specialists
  • (2) provides a preview of your objective of defining circulating factors that may be identified in either blood or the less pleasant acquisition of CSF
  • (3) and that it includes markers that may or may not require highly skilled teams to measure.  This could provide a construct for the reader in assessing the potential general value of further exploration of each approach beyond the sensitivity and specificity identified in existing studies that are generally small and/or without validation on independent data sets,

A recent reference of a novel approach including machine learning that may be discussed (i realize this is rapidly moving field):  Mathios D, Niknafs N, Annapragada AV, Bobeff EJ, Chiao EJ, Boyapati K, Boyapati K, Short S, Bartolomucci AL, Cristiano S, Koul S, Vulpescu NA, Ferreira L, Medina JE, Bruhm DC, Adleff V, Podstawka M, StanisÅ‚awska P, Park CK, Huang J, Gallia GL, Brem H, Mukherjee D, Caplan JM, Weingart J, Jackson CM, Lim M, Phallen J, Scharpf RB, Velculescu VE. Detection of brain cancer using genome-wide cell-free DNA fragmentomes. Cancer Discov. 2025 Apr 29. doi: 10.1158/2159-8290.CD-25-0074. Epub ahead of print. PMID: 40299319.

Thank you for the input we have now added the below overview in the conclusion on page 21, lines 722-728 as follows: “Imaging alone remains unreliable for distinguishing true progression from treatment effects, highlighting the need for adjunct approaches. Liquid biopsy markers vary in their technical demands, from easily measurable analytes to those requiring specialized platforms, and assessing their clinical utility requires more than sensitivity or specificity alone. Recent advances such as genome-wide fragmentomic analysis using machine learning[1] demonstrate the potential of scalable and innovative approaches in this space.”

  • One sentence jumped out as confusing--

Liquid biopsies comprise the detection and quantification of tumour related content in bio- 69
fluids that is released by tumours, such as circulating tumour cells (CTCs), circulating 70
tumour DNA (ctDNA), circulating tumour microRNA (miRNA), extracellular vesicles 71
(EVs) and nucleosomes or in response to tumours, all of which can provide critical insights 72
into GBM progression and recurrence.

might be changed to something similar to:

Liquid biopsies comprise the detection and quantification of tumour related content in bio-
fluids that is released by tumours or in response to the presence of tumor, such as circulating tumour cells (CTCs), circulating tumour DNA (ctDNA), circulating tumour microRNA (miRNA), extracellular vesicles (EVs) and nucleosomes or in response to tumours, all of which can provide critical insights into GBM progression and recurrence.

Thank you, we agree and have made this edit on page 3, line 88-92.

Reviewer 2 Report

Comments and Suggestions for Authors

Thank you for providing me with the opportunity to review this important review article. This experience has been highly educational and informative. I submit the following review comments as suggestions for improvement, which I hope will be useful for your consideration when needed.

For each testing modality presented in the tables, please include the minimum required input DNA quantity and analytical sensitivity in addition to patient numbers.

While CSF is mentioned in the manuscript, it appears to represent a fundamentally different analytical approach. I recommend creating a separate section to discuss CSF analysis more thoroughly and comprehensively.

The discussion regarding the differentiation of ATRX and TP53 alterations from clonal hematopoiesis should be expanded and addressed more rigorously.

SOX2 and GFAP appear to represent different analytical approaches compared to ctDNA-based liquid biopsy. A more detailed explanation would enhance reader comprehension of these distinct methodologies.

Given that temozolomide treatment significantly alters the DNA landscape, more detailed discussion of this important clinical consideration would be beneficial.

To enhance the practical value of this review for clinicians, consider including discussion of the utility in scenarios where biopsy is technically challenging, such as brainstem gliomas.

Chromosomal alterations and EGFR structural variants represent analytical challenges for next-generation sequencing platforms with potential for inconclusive results. Detailed discussion of these technical limitations would increase the clinical utility of this review.

Where applicable, please ensure that discussions are based on the most recent WHO classification system.

Expansion of the discussion regarding pediatric cases would be greatly appreciated and would enhance the comprehensiveness of this review.

Author Response

Reviewer 2

Thank you for providing me with the opportunity to review this important review article. This experience has been highly educational and informative. I submit the following review comments as suggestions for improvement, which I hope will be useful for your consideration when needed.

  • For each testing modality presented in the tables, please include the minimum required input DNA quantity and analytical sensitivity in addition to patient numbers.

We agree with the reviewer the minimum input DNA would be valuable to add so we have added another column to the right of patient numbers for minimum input DNA on page 8-10 where specified in Table 2. However, the minimum possible input is not normally reported, so we have added “not specified” where it was not reported. Analytical sensitivity where specified has been added in the “Findings” column of Table 2. Patient numbers analysed should always be provided, remain listed in column 2 of each table. Where required the text has been updated to prevent repetition.

  • While CSF is mentioned in the manuscript, it appears to represent a fundamentally different analytical approach. I recommend creating a separate section to discuss CSF analysis morethoroughly and comprehensively.

To address this comment, we have revised the relevant paragraphs and added on page 3 lines 134-152 and lines 200-201 on page 6 to more clearly highlight the unique aspects of CSF sampling and analysis, including its higher analyte concentrations, and procedural considerations. Specific studies demonstrating its utility in GBM are listed within the tables clearly.

We have also added the essential considerations required when designing AI/ML algorithms for the data derived from sources like CSF which is fundamentally different than other biofluids (line 690-695 on page 20) as follows “Importantly, while integrating findings across biofluids provides a broad perspective, future studies should also recognise the distinct analytical considerations of each matrix. For example, CSF offers higher CNS-specific biomarker abundance, reduced systemic noise, and unique analyte profiles, but requires low-input assay adaptations and safeguards against blood contamination. Incorporating such matrix-specific knowledge into AI/ML-driven multi-biofluid frameworks may accelerate clinical translation” We believe this approach preserves clarity while integrating CSF analysis within the broader discussion of circulating biomarkers provides a more cohesive narrative.

  • The discussion regarding the differentiation of ATRX and TP53 alterations from clonal haematopoiesis should be expanded and addressed more rigorously.

We have now added a discussion on page 6 lines 216-228 as follows: “It is important to emphasize that mutation detection in cf/ctDNA, in particular detection of rare alleles, may be affected by biological background noise. Clonal hematopoiesis (CH) is an event triggered by accumulation of somatic mutations during development of hemopoietic cells followed by the outgrowth of genetically more diverse clones. This is a normal process with increasing accumulation of clones with age. Critically, if certain variants in DNA damage response proteins including ATRX and TP53 arise, outgrowth of clones with these mutations (and others in the same cell) is favoured due to impaired DNA repair[2,3]. This means mutations that are generally associated with cancer may also accumulate and be detectable in the non-cancerous blood cell derived component of cfDNA of healthy, especially older individuals. Thus, CH affects the interpretation of liquid biopsy mutation data for rare alleles in GBM. This must be kept in mind and should be controlled for by independent methods of determining ctDNA proportion or the presence of CH mutations in the immune cell population of a patient sample.”

  • SOX2 and GFAP appear to represent different analytical approaches compared to ctDNA-based liquid biopsy. A more detailed explanation would enhance reader comprehension of these distinct methodologies.

We have addressed the following comment on page 4 lines 154-157 as follows “Notably, markers like SOX2 and GFAP are detected via protein expression using immunocytochemistry or flow cytometry, whereas transcript analysis methods such as RT-PCR target mRNA expression, distinguishing these approaches technically.

  • Given that temozolomide treatment significantly alters the DNA landscape, more detailed discussion of this important clinical consideration would be beneficial.

Thankyou for the comment, we have now added a brief discussion on Page 7, line 297-305 as follows “TMZ, a DNA alkylating agent, adds methyl groups primarily to the O6 position of guanine, causing lethal DNA damage. In normal scenarios, MGMT repairs this damage, while its promoter methylation or mutation impairs MGMT function and increases tumour sensitivity to TMZ. Importantly, TMZ treatment also induces substantial changes in the tumour DNA landscape, including increased mutational burden and epigenetic remodelling. These therapy induced alterations can affect the detection and interpretation of ctDNA and methylation biomarkers, potentially complicating longitudinal monitoring. Therefore, understanding TMZ’s impact on liquid biopsy profiles is essential for accurately assessing treatment response and resistance.”

  • To enhance the practical value of this review for clinicians, consider including discussion of the utility in scenarios where biopsy is technically challenging, such as brainstem gliomas.

Thank you for the comment, we have now added on page 20 lines 704-711 as follows “In the past, clinicians may have ignored the liquid biopsy literature, due to the difficulties in reliable detection of rare events, but as more data is presented, and as techniques and methodologies improve, liquid biopsies, particularly for patients with brain cancer (primary and metastatic), offer a compelling advance in brain cancer therapy. Increasingly, where tissue access is contra-indicated (eg. brain stem gliomas, tumours in eloquent areas), not recommended (eg. patients with co-morbidities or frailty), or inconvenient (eg. CSF monitoring), liquid biopsies, alone or in conjunction with tra-ditional imaging, may represent the best way to aid in the management of these patients.

  • Chromosomal alterations and EGFR structural variants represent analytical challenges for next-generation sequencing platforms with potential for inconclusive results. Detailed discussion of these technical limitations would increase the clinical utility of this review.

We agree with this comment. We added some discussion and suggest that other approaches such as ddPCR based for EGFRvIII may be superior in the context on Page 6 lines 214-215

  • Where applicable, please ensure that discussions are based on the most recent WHO classification system.

This is an important comment but the clarification whether the most recent WHO classification was used is not always included. While it would be clear that publications prior the change would not use the new classification, many newer reports may still include patient samples obtained prior. We have now added a sentence in the Conclusion that deals with this important aspect on Page 20, lines 711-712 as follows “Although this review cannot always confirm use of the latest WHO classification, its findings remain relevant.

  • Expansion of the discussion regarding pediatric cases would be greatly appreciated and would enhance the comprehensiveness of this review.

We agree with the reviewer that paediatric gliomas are highly relevant and often have a different biology. However, we feel to do this topic justice (go into it comprehensively enough) would go beyond and is not within the scope of this review. Instead, we are now referring the reader to recent reviews in this area on Page 11, line 415-418 as follows “While this review focussed on adult glioma, the analysis of liquid biopsy in paediatric glioma with often different biology faces other problems such as sample volume, which has recently been in detail reviewed by others[4-7].”

  1. Mathios, D.; Niknafs, N.; Annapragada, A.V.; Bobeff, E.J.; Chiao, E.J.; Boyapati, K.; Boyapati, K.; Short, S.; Bartolomucci, A.L.; Cristiano, S.; et al. Detection of Brain Cancer Using Genome-wide Cell-free DNA Fragmentomes. Cancer Discov 2025, 15, 1593-1608, doi:10.1158/2159-8290.Cd-25-0074.
  2. Chan, H.T.; Chin, Y.M.; Nakamura, Y.; Low, S.K. Clonal Hematopoiesis in Liquid Biopsy: From Biological Noise to Valuable Clinical Implications. Cancers (Basel) 2020, 12, doi:10.3390/cancers12082277.
  3. Jensen, K.; Konnick, E.Q.; Schweizer, M.T.; Sokolova, A.O.; Grivas, P.; Cheng, H.H.; Klemfuss, N.M.; Beightol, M.; Yu, E.Y.; Nelson, P.S.; et al. Association of Clonal Hematopoiesis in DNA Repair Genes With Prostate Cancer Plasma Cell-free DNA Testing Interference. JAMA Oncol 2021, 7, 107-110, doi:10.1001/jamaoncol.2020.5161.
  4. Biegel, J.A. Meeting the high expectations for liquid biopsy assays for pediatric brain tumors: Progress and challenges. Neuro Oncol 2022, 24, 1364-1365, doi:10.1093/neuonc/noac083.
  5. Tang, K.; Gardner, S.; Snuderl, M. The Role of Liquid Biopsies in Pediatric Brain Tumors. J Neuropathol Exp Neurol 2020, 79, 934-940, doi:10.1093/jnen/nlaa068.
  6. Tripathy, A.; John, V.; Wadden, J.; Kong, S.; Sharba, S.; Koschmann, C. Liquid biopsy in pediatric brain tumors. Front Genet 2022, 13, 1114762, doi:10.3389/fgene.2022.1114762.
  7. Bounajem, M.T.; Karsy, M.; Jensen, R.L. Liquid biopsies for the diagnosis and surveillance of primary pediatric central nervous system tumors: a review for practicing neurosurgeons. Neurosurg Focus 2020, 48, E8, doi:10.3171/2019.9.Focus19712.